# Provable Benefit of Orthogonal Initialization in Optimizing Deep Linear Networks

**Wei Hu**
Princeton University
huwei@cs.princeton.edu

**Lechao Xiao**
Google Brain
xlc@google.com

**Jeffrey Pennington**
Google Brain
jpennin@google.com

## Abstract

The selection of initial parameter values for gradient-based optimization of deep neural networks is one of the most impactful hyperparameter choices in deep learning systems, affecting both convergence times and model performance. Yet despite significant empirical and theoretical analysis, relatively little has been proved about the concrete effects of different initialization schemes. In this work, we analyze the effect of initialization in deep linear networks, and provide for the first time a rigorous proof that drawing the initial weights from the orthogonal group speeds up convergence relative to the standard Gaussian initialization with iid weights. We show that for deep networks, the width needed for efficient convergence to a global minimum with orthogonal initializations is independent of the depth, whereas the width needed for efficient convergence with Gaussian initializations scales linearly in the depth. Our results demonstrate how the benefits of a good initialization can persist throughout learning, suggesting an explanation for the recent empirical successes found by initializing very deep non-linear networks according to the principle of *dynamical isometry*.

## 1 Introduction

Through their myriad successful applications across a wide range of disciplines, it is now well established that deep neural networks possess an unprecedented ability to model complex real-world datasets, and in many cases they can do so with minimal overfitting. Indeed, the list of practical achievements of deep learning has grown at an astonishing rate, and includes models capable of human-level performance in tasks such as image recognition (Krizhevsky et al., 2012), speech recognition (Hinton et al., 2012), and machine translation (Wu et al., 2016).

Yet to each of these deep learning triumphs corresponds a large engineering effort to produce such a high-performing model. Part of the practical difficulty in designing good models stems from a proliferation of hyperparameters and a poor understanding of the general guidelines for their selection. Given a candidate network architecture, some of the most impactful hyperparameters are those governing the choice of the model's initial weights. Although considerable study has been devoted to the selection of initial weights, relatively little has been proved about how these choices affect important quantities such as rate of convergence of gradient descent.

In this work, we examine the effect of initialization on the rate of convergence of gradient descent in deep linear networks. We provide for the first time a rigorous proof that drawing the initial weights from the orthogonal group speeds up convergence relative to the standard Gaussian initialization with iid weights. In particular, we show that for deep networks, the width needed for efficient convergence for orthogonal initializations is independent of the depth, whereas the width needed for efficient convergence of Gaussian networks scales linearly in the depth.

Orthogonal weight initializations have been the subject of a significant amount of prior theoretical and empirical investigation. For example, in a line of work focusing on *dynamical isometry*, it was found that orthogonal weights can speed up convergence for deep linear networks (Saxe et al., 2014; Advani & Saxe, 2017) and for deep non-linear networks (Pennington et al., 2018; Xiao et al., 2018; Gilboa et al., 2019; Chen et al., 2018; Pennington et al., 2017; Tarnowski et al., 2019; Ling & Qiu, 2019) when they operate in the linear regime. In the context of recurrent neural networks, orthogonality can help improve the system's stability. A main limitation of prior work is that it

has focused almost exclusively on model's properties at initialization. In contrast, our analysis focuses on the benefit of orthogonal initialization on the entire training process, thereby establishing a provable benefit for optimization.

The paper is organized as follows. After reviewing related work in Section 2 and establishing some preliminaries in Section 3, we present our main positive result on efficient convergence from orthogonal initialization in Section 4. In Section 5, we show that Gaussian initialization leads to exponentially long convergence time if the width is too small compared with the depth. In Section 6, we perform experiments to support our theoretical results.

## 2 RELATED WORK

**Deep linear networks.** Despite the simplicity of their input-output maps, deep linear networks define high-dimensional non-convex optimization landscapes whose properties closely reflect those of their non-linear counterparts. For this reason, deep linear networks have been the subject of extensive theoretical analysis. A line of work (Kawaguchi, 2016; Hardt & Ma, 2016; Lu & Kawaguchi, 2017; Yun et al., 2017; Zhou & Liang, 2018; Laurent & von Brecht, 2018) studied the landscape properties of deep linear networks. Although it was established that all local minima are global under certain assumptions, these properties alone are still not sufficient to guarantee global convergence or to provide a concrete rate of convergence for gradient-based optimization algorithms.

Another line of work directly analyzed the trajectory taken by gradient descent and established conditions that guarantee convergence to global minimum (Bartlett et al., 2018; Arora et al., 2018; Du & Hu, 2019). Most relevant to our work is the result of Du & Hu (2019), which shows that if the width of hidden layers is larger than the depth, gradient descent with Gaussian initialization can efficiently converge to a global minimum. Our result establishes that for Gaussian initialization, this linear dependence between width and depth is necessary, while for orthogonal initialization, the width can be independent of depth. Our negative result for Gaussian initialization also significantly generalizes the result of Shamir (2018), who proved a similar negative result for 1-dimensional linear networks.

**Orthogonal weight initializations.** Orthogonal weight initializations have also found significant success in non-linear networks. In the context of feedforward models, the spectral properties of a network's input-output Jacobian have been empirically linked to convergence speed (Saxe et al., 2014; Pennington et al., 2017; 2018; Xiao et al., 2018). It was found that when this spectrum concentrates around 1 at initialization, a property dubbed *dynamical isometry*, convergence times improved by orders of magnitude. The conditions for attaining dynamical isometry in the infinite-width limit were established by Pennington et al. (2017; 2018) and basically require that input-output map to be approximately linear and for the weight matrices to be orthogonal. Therefore the training time benefits of dynamical isometry are likely rooted in the benefits of orthogonality for deep linear networks, which we establish in this work.

Orthogonal matrices are also frequently used in the context of recurrent neural networks, for which the stability of the state-to-state transition operator is determined by the spectrum of its Jacobian (Haber & Ruthotto, 2017; Laurent & von Brecht, 2016). Orthogonal matrices can improve the conditioning, leading to an ability to learn over long time horizons (Le et al., 2015; Henaff et al., 2016; Chen et al., 2018; Gilboa et al., 2019). While the benefits of orthogonality can be quite large at initialization, little is known about whether or in what contexts these benefits persist during training, a scenario that has lead to the development of efficient methods of constraining the optimization to the orthogonal group (Wisdom et al., 2016; Vorontsov et al., 2017; Mhammedi et al., 2017). Although we do not study the recurrent setting in this work, an extension of our analysis might help determine when orthogonality is beneficial in that setting.

## 3 PRELIMINARIES

### 3.1 NOTATION

Let $[n] = \{1, 2, \ldots, n\}$. Denote by $I_d$ the $d \times d$ identity matrix, and by $I$ an identity matrix when its dimension is clear from context. Denote by $\mathcal{N}(\mu, \sigma^2)$ the Gaussian distribution with mean $\mu$ and variance $\sigma^2$, and by $\chi_k^2$ the chi-squared distribution with $k$ degrees of freedom.

Denote by $\|\cdot\|$ the $\ell_2$ norm of a vector or the spectral norm of a matrix. Denote by $\|\cdot\|_F$ the Frobenius norm of a matrix. For a symmetric matrix $A$, let $\lambda_{\max}(A)$ and $\lambda_{\min}(A)$ be its maximum and minimum eigenvalues, and let $\lambda_i(A)$ be its $i$-th largest eigenvalue. For a matrix $B \in \mathbb{R}^{m \times n}$, let $\sigma_i(B)$ be its $i$-th largest singular value ($i = 1, 2, \ldots, \min\{m, n\}$), and let $\sigma_{\max}(B) = \sigma_1(B)$, $\sigma_{\min}(B) = \sigma_{\min\{m,n\}}(B)$. Denote by $\text{vec}(A)$ be the vectorization of a matrix $A$ in column-first order. The Kronecker product between two matrices $A \in \mathbb{R}^{m_1 \times n_1}$ and $B \in \mathbb{R}^{m_2 \times n_2}$ is defined as

$$A \otimes B = \begin{pmatrix} a_{1,1}B & \cdots & a_{1,n_1}B \\ \vdots & \ddots & \vdots \\ a_{m_1,1}B & \cdots & a_{m_1,n_1}B \end{pmatrix} \in \mathbb{R}^{m_1 m_2 \times n_1 n_2},$$

where $a_{i,j}$ is the element in the $(i, j)$-th entry of $A$.

We use the standard $O(\cdot)$, $\Omega(\cdot)$ and $\Theta(\cdot)$ notation to hide universal constant factors. We also use $C$ to represent a sufficiently large universal constant whose specific value can differ from line to line.

### 3.2 PROBLEM SETUP

Suppose that there are $n$ training examples $\{(x_k, y_k)\}_{k=1}^n \subset \mathbb{R}^{d_x} \times \mathbb{R}^{d_y}$. Denote by $X = (x_1, \ldots, x_n) \in \mathbb{R}^{d_x \times n}$ the input data matrix and by $Y = (y_1, \ldots, y_n) \in \mathbb{R}^{d_y \times n}$ the target matrix. Consider an $L$-layer linear neural network with weight matrices $W_1, \ldots, W_L$, which given an input $x \in \mathbb{R}^{d_x}$ computes

$$f(x; W_1, \ldots, W_L) = \alpha W_L W_{L-1} \cdots W_1 x, \tag{1}$$

where $W_i \in \mathbb{R}^{d_i \times d_{i-1}} (i = 1, \ldots, L)$, $d_0 = d_x$, $d_L = d_y$, and $\alpha$ is a normalization constant which will be specified later according to the initialization scheme. We study the problem of training the deep linear network by minimizing the $\ell_2$ loss over training data:

$$\ell(W_1, \ldots, W_L) = \frac{1}{2} \sum_{k=1}^n \|f(x_k; W_1, \ldots, W_L) - y_k\|^2 = \frac{1}{2} \|\alpha W_L \cdots W_1 X - Y\|_F^2. \tag{2}$$

The algorithm we consider to minimize the objective (2) is gradient descent with random initialization, which first randomly samples the initial weight matrices $\{W_i(0)\}_{i=1}^L$ from a certain distribution, and then updates the weights using gradient descent: for time $t = 0, 1, 2, \ldots$,

$$W_i(t+1) = W_i(t) - \eta \frac{\partial \ell}{\partial W_i}(W_1(t), \ldots, W_L(t)), \qquad i \in [L], \tag{3}$$

where $\eta > 0$ is the learning rate.

For convenience, we denote $W_{j:i} = W_j W_{j-1} \cdots W_i (1 \le i \le j \le L)$ and $W_{i-1:i} = I (i \in [L])$. The time index $t$ is used on any variable that depends on $W_1, \ldots, W_L$ to represent its value at time $t$, e.g., $W_{j:i}(t) = W_j(t) \cdots W_i(t)$, $\ell(t) = \ell(W_1(t), \ldots, W_L(t))$, etc.

## 4 EFFICIENT CONVERGENCE USING ORTHOGONAL INITIALIZATION

In this section we present our main positive result for orthogonal initialization. We show that orthogonal initialization enables efficient convergence of gradient descent to a global minimum provided that the hidden width is not too small.

In order to properly define orthogonal weights, we let the widths of all hidden layers be equal: $d_1 = d_2 = \cdots = d_{L-1} = m$, and let $m \ge \max\{d_x, d_y\}$. Note that all intermediate matrices

$W_2, \dots, W_{L-1}$ are $m \times m$ square matrices, and $W_1 \in \mathbb{R}^{m \times d_x}, W_L \in \mathbb{R}^{d_y \times m}$. We sample each initial weight matrix $W_i(0)$ independently from a uniform distribution over scaled orthogonal matrices satisfying

$$
\begin{aligned}
W_1^\top(0)W_1(0) &= mI_{d_x}, \\
W_i^\top(0)W_i(0) = W_i(0)W_i^\top(0) &= mI_m, \qquad 2 \le i \le L-1, \\
W_L(0)W_L^\top(0) &= mI_{d_y}.
\end{aligned}
\qquad (4)
$$

In accordance with such initialization, the scaling factor $\alpha$ in (1) is set as $\alpha = \frac{1}{\sqrt{m^{L-1}d_y}}$, which ensures $\mathbb{E}\left[\|f(x; W_L(0), \dots, W_1(0))\|^2\right] = \|x\|^2$ for any $x \in \mathbb{R}^{d_x}$.[1] The same scaling factor was adopted in Du & Hu (2019), which preserves the expectation of the squared $\ell_2$ norm of any input.

Let $W^* \in \arg\min_{W \in \mathbb{R}^{d_y \times d_x}} \|WX - Y\|_F$ and $\ell^* = \frac{1}{2}\|W^*X - Y\|_F^2$. Then $\ell^*$ is the minimum value for the objective (2). Denote $r = \operatorname{rank}(X)$, $\kappa = \frac{\lambda_{\max}(X^\top X)}{\lambda_r(X^\top X)}$, and $\tilde{r} = \frac{\|X\|_F^2}{\|X\|^2}$.[2] Our main theorem in this section is the following:

**Theorem 4.1.** *Suppose*

$$
m \ge C \cdot \tilde{r}\kappa^2 \left( d_y(1 + \|W^*\|^2) + \log(r/\delta) \right) \ and \ m \ge d_x,
\qquad (5)
$$

*for some $\delta \in (0, 1)$ and a sufficiently large universal constant $C > 0$. Set the learning rate $\eta \le \frac{d_y}{2L\|X\|^2}$. Then with probability at least $1 - \delta$ over the random initialization, we have*

$$
\ell(0) - \ell^* \le O\left( 1 + \frac{\log(r/\delta)}{d_y} + \|W^*\|^2 \right) \|X\|_F^2,
$$

$$
\ell(t) - \ell^* \le \left( 1 - \frac{1}{2}\eta L\lambda_r(X^\top X)/d_y \right)^t (\ell(0) - \ell^*), \quad t = 0, 1, 2, \dots,
$$

*where $\ell(t)$ is the objective value at iteration $t$.*

Notably, in Theorem 4.1, the width $m$ need not depend on the depth $L$. This is in sharp contrast with the result of Du & Hu (2019) for Gaussian initialization, which requires $m \ge \tilde{\Omega}(Lr\kappa^3 d_y)$. It turns out that a near-linear dependence between $m$ and $L$ is necessary for Gaussian initialization to have efficient convergence, as we will show in Section 5. Therefore the requirement in Du & Hu (2019) is nearly tight in terms of the dependence on $L$. These results together rigorously establish the benefit of orthogonal initialization in optimizing very deep linear networks.

If we set the learning rate optimally according to Theorem 4.1 to $\eta = \Theta(\frac{d_y}{L\|X\|^2})$, we obtain that $\ell(t) - \ell^*$ decreases by a ratio of $1 - \Theta(\kappa^{-1})$ after every iteration. This matches the convergence rate of gradient descent on the (1-layer) linear regression problem $\min_{W \in \mathbb{R}^{d_y \times d_x}} \frac{1}{2}\|WX - Y\|_F^2$.

## 4.1 Proof of Theorem 4.1

The proof uses the high-level framework from Du & Hu (2019), which tracks the evolution of the network's output during optimization. This evolution is closely related to a time-varying positive semidefinite (PSD) matrix (defined in (7)), and the proof relies on carefully upper and lower bounding the eigenvalues of this matrix throughout training, which in turn implies the desired convergence result.

First, we can make the following simplifying assumption *without loss of generality*. See Appendix B in Du & Hu (2019) for justification.

**Assumption 4.1.** *(Without loss of generality)* $X \in \mathbb{R}^{d_x \times r}$, $\operatorname{rank}(X) = r$, $Y = W^*X$, and $\ell^* = 0$.

---

[1]We have $\mathbb{E}\left[\|f(x; W_L(0), \dots, W_1(0))\|^2\right] = \alpha^2 \mathbb{E}\left[x^\top W_1^\top(0) \cdots W_L^\top(0)W_L(0) \cdots W_1(0)x\right]$. Note that by our choice (4) we have $\mathbb{E}\left[W_L^\top(0)W_L(0)\right] = d_y I_m$ and $W_i^\top(0)W_i(0) = mI$ ($1 \le i \le L-1$), so we have $\mathbb{E}\left[\|f(x; W_L(0), \dots, W_1(0))\|^2\right] = \alpha^2 m^{L-1}d_y \|x\|^2 = \|x\|^2$.

[2]$\tilde{r}$ is known as the *stable rank* of $X$, which is always no more than the rank.

Now we briefly review Du & Hu (2019)'s framework. The key idea is to look at the network's output, defined as

$$U = \alpha W_{L:1}X \in \mathbb{R}^{d_y \times n}.$$

We also write $U(t) = \alpha W_{L:1}(t)X$ as the output at time $t$. Note that $\ell(t) = \frac{1}{2}\|U(t) - Y\|_F^2$. According to the gradient descent update rule, we write

$$W_{L:1}(t+1) = \prod_i \left(W_i(t) - \eta\frac{\partial\ell}{\partial W_i}(t)\right) = W_{L:1}(t) - \sum_{i=1}^{L}\eta W_{L:i+1}(t)\frac{\partial\ell}{\partial W_i}(t)W_{i-1:1}(t) + E(t),$$

where $E(t)$ contains all the high-order terms (i.e., those with $\eta^2$ or higher). With this definition, the evolution of $U(t)$ can be written as the following equation:

$$\text{vec}\left(U(t+1) - U(t)\right) = -\eta P(t) \cdot \text{vec}\left(U(t) - Y\right) + \alpha \cdot \text{vec}\left(E(t)X\right), \tag{6}$$

where

$$P(t) = \alpha^2 \sum_{i=1}^{L}\left[\left((W_{i-1:1}(t)X)^\top (W_{i-1:1}(t)X)\right) \otimes \left(W_{L:i+1}(t)W_{L:i+1}^\top(t)\right)\right]. \tag{7}$$

Notice that $P(t)$ is always PSD since it is the sum of $L$ PSD matrices. Therefore, in order to establish convergence, we only need to (i) show that the higher-order term $E(t)$ is small and (ii) prove upper and lower bounds on $P(t)$'s eigenvalues. For the second task, it suffices to control the singular values of $W_{i-1:1}(t)$ and $W_{L:i+1}(t)$ ($i \in [L]$).[3] Under orthogonal initialization, these matrices are perfectly isometric at initialization, and we will show that they stay close to isometry during training, thus enabling efficient convergence.

The following lemma summarizes some properties at initialization.

**Lemma 4.2.** *At initialization, we have*

$$\sigma_{\max}(W_{j:i}(0)) = \sigma_{\min}(W_{j:i}(0)) = m^{\frac{j-i+1}{2}}, \quad \forall 1 \leq i \leq j \leq L, (i,j) \neq (1,L). \tag{8}$$

*Furthermore, with probability at least $1 - \delta$, the loss at initialization satisfies*

$$\ell(0) \leq O\left(1 + \frac{\log(r/\delta)}{d_y} + \|W^*\|^2\right)\|X\|_F^2. \tag{9}$$

*Proof sketch.* The spectral property (8) follows directly from (4).

To prove (9), we essentially need to upper bound the magnitude of the network's initial output. This turns out to be equivalent to studying the magnitude of the projection of a vector onto a random low-dimensional subspace, which we can bound using standard concentration inequalities. The details are given in Appendix A.1. $\square$

Now we proceed to prove Theorem 4.1. We define $B = O\left(1 + \frac{\log(r/\delta)}{d_y} + \|W^*\|^2\right)\|X\|_F^2$ which is the upper bound on $\ell(0)$ from (9). Conditioned on (9) being satisfied, we will use induction on $t$ to prove the following three properties $\mathcal{A}(t)$, $\mathcal{B}(t)$ and $\mathcal{C}(t)$ for all $t = 0, 1, \ldots$:

- $\mathcal{A}(t)$: $\ell(t) \leq \left(1 - \frac{1}{2}\eta L\sigma_{\min}^2(X)/d_y\right)^t \ell(0) \leq \left(1 - \frac{1}{2}\eta L\sigma_{\min}^2(X)/d_y\right)^t B$.
- $\mathcal{B}(t)$: $\sigma_{\max}(W_{j:i}(t)) \leq 1.1 m^{\frac{j-i+1}{2}}, \sigma_{\min}(W_{j:i}(t)) \geq 0.9 m^{\frac{j-i+1}{2}}, \quad \forall 1 \leq i \leq j \leq L, (i,j) \neq (1,L)$.
- $\mathcal{C}(t)$: $\|W_i(t) - W_i(0)\|_F \leq \frac{8\sqrt{Bd_y}\|X\|}{L\sigma_{\min}^2(X)}, \quad \forall 1 \leq i \leq L$.

$\mathcal{A}(0)$ and $\mathcal{B}(0)$ are true according to Lemma 4.2, and $\mathcal{C}(0)$ is trivially true. In order to prove $\mathcal{A}(t)$, $\mathcal{B}(t)$ and $\mathcal{C}(t)$ for all $t$, we will prove the following claims for all $t \geq 0$:

---

[3]Note that for symmetric matrices $A$ and $B$, the set of eigenvalues of $A \otimes B$ is the set of products of an eigenvalue of $A$ and an eigenvalue of $B$.

**Claim 4.3.** $\mathcal{A}(0), \ldots, \mathcal{A}(t), \mathcal{B}(0), \ldots, \mathcal{B}(t) \implies \mathcal{C}(t+1)$.

**Claim 4.4.** $\mathcal{C}(t) \implies \mathcal{B}(t)$.

**Claim 4.5.** $\mathcal{A}(t), \mathcal{B}(t) \implies \mathcal{A}(t+1)$.

The proofs of these claims are given in Appendix A. Notice that we finish the proof of Theorem 4.1 once we prove $\mathcal{A}(t)$ for all $t \geq 0$.

# 5 EXPONENTIAL CURSE OF GAUSSIAN INITIALIZATION

In this section, we show that gradient descent with Gaussian random initialization necessarily suffers from a running time that scales exponentially with the depth of the network, unless the width becomes nearly linear in the depth. Since we mostly focus on the dependence of width and running time on depth, we will assume the depth $L$ to be sufficiently large.

Recall that we want to minimize the objective $\ell(W_1, \ldots, W_L) = \frac{1}{2} \|\alpha W_{L:1} X - Y\|_F^2$ by gradient descent. We assume $Y = W^* X$ for some $W^* \in \mathbb{R}^{d_y \times d_x}$, so that the optimal objective value is 0. For convenience, we assume $\|X\|_F = \Theta(1)$ and $\|Y\|_F = \Theta(1)$.

Suppose that at layer $i \in [L]$, every entry of $W_i(0)$ is sampled from $\mathcal{N}(0, \sigma_i^2)$, and all weights in the network are independent. We set the scaling factor $\alpha$ such that the initial output of the network does not blow up exponentially (in expectation):

$$\mathbb{E}\left[\|f(x; W_1(0), \ldots, W_L(0))\|^2\right] \leq L^{O(1)} \cdot \|x\|^2, \quad \forall x \in \mathbb{R}^{d_x}. \tag{10}$$

Note that $\mathbb{E}\left[\|f(x; W_1(0), \ldots, W_L(0))\|^2\right] = \alpha^2 \prod_{i=1}^{L}(d_i \sigma_i^2) \|x\|^2$. Thus (10) means

$$\alpha^2 \prod_{i=1}^{L}(d_i \sigma_i^2) \leq L^{O(1)}.$$

We also assume that the magnitude of initialization at each layer cannot vanish with depth:

$$d_i \sigma_i^2 \geq \frac{1}{L^{O(1)}}, \quad \forall i \in [L]. \tag{11}$$

Note that the assumptions (10) and (11) are just sanity checks to rule out the obvious pathological cases – they are easily satisfied by all the commonly used initialization schemes in practice.

Now we formally state our main theorem in this section.

**Theorem 5.1.** *Suppose* $\max\{d_0, d_1, \ldots, d_L\} \leq O(L^{1-\gamma})$ *for some universal constant* $0 < \gamma \leq 1$. *Then there exists a universal constant* $c > 0$ *such that, if gradient descent is run with learning rate* $\eta \leq e^{cL^\gamma}$, *then with probability at least* $0.9$ *over the random initialization, for the first* $e^{\Omega(L^\gamma)}$ *iterations, the objective value is stuck between* $0.4 \|Y\|_F^2$ *and* $0.6 \|Y\|_F^2$.

Theorem 5.1 establishes that efficient convergence from Gaussian initialization is impossible for large depth unless the width becomes nearly linear in depth. This nearly linear dependence is the best we can hope for, since Du & Hu (2019) proved a positive result when the width is larger than linear in depth. Therefore, a phase transition from untrainable to trainable happens at the point when the width and depth has a nearly linear relation. Furthermore, Theorem 5.1 generalizes the result of Shamir (2018), which only treats the special case of $d_0 = \cdots = d_L = 1$.

## 5.1 PROOF OF THEOREM 5.1

For convenience, we define a scaled version of $W_i$: let $A_i = W_i/(\sqrt{d_i}\sigma_i)$ and $\beta = \alpha \prod_{i=1}^{L}(\sqrt{d_i}\sigma_i)$. Then we know $\beta \leq L^{O(1)}$ and $\alpha W_{L:1} = \beta A_{L:1}$, where $A_{j:i} = A_j \cdots A_i$.

We first give a simple upper bound on $\|A_{j:i}(0)\|$ for all $1 \leq i \leq j \leq L$.

**Lemma 5.2.** *With probability at least* $1 - \delta$, *we have* $\|A_{j:i}(0)\| \leq O\left(\frac{L^3}{\delta}\right)$ *for all* $1 \leq i \leq j \leq L$.

The proof of Lemma 5.2 is given in Appendix B.1. It simply uses Markov inequality and union bound.

Furthermore, a key property at initialization is that if $j - i$ is large enough, $\|A_{j:i}(0)\|$ will become exponentially small.

**Lemma 5.3.** *With probability at least $1 - e^{-\Omega(L^\gamma)}$, for all $1 \le i \le j \le L$ such that $j - i \ge \frac{L}{10}$, we have $\|A_{j:i}(0)\| \le e^{-\Omega(L^\gamma)}$.*

*Proof.* We first consider a fixed pair $(i, j)$ such that $j - i \ge \frac{L}{10}$. In order to bound $\|A_{j:i}(0)\|$, we first take an arbitrary unit vector $v \in \mathbb{R}^{d_{i-1}}$ and bound $\|A_{j:i}(0)v\|$. We can write $\|A_{j:i}(0)v\|^2 = \prod_{k=i}^{j} Z_k$, where $Z_k = \frac{\|A_{k:i}(0)v\|^2}{\|A_{k-1:i}(0)v\|^2}$. Note that for any nonzero $v' \in \mathbb{R}^{d_{k-1}}$ independent of $A_k(0)$, the distribution of $d_k \cdot \frac{\|A_k(0)v'\|^2}{\|v'\|^2}$ is $\chi^2_{d_k}$. Therefore, $Z_i, \ldots, Z_j$ are independent, and $d_k Z_k \sim \chi^2_{d_k}$ $(k = i, i+1, \ldots, j)$. Recall the expression for the moments of chi-squared random variables: $\mathbb{E}\left[Z_k^\lambda\right] = \frac{2^\lambda \Gamma(d_k/2 + \lambda)}{d_k^\lambda \Gamma(d_k/2)}$ $(\forall \lambda > 0)$. Taking $\lambda = \frac{1}{2}$ and using the bound $\frac{\Gamma(a + \frac{1}{2})}{\Gamma(a)} \le \sqrt{a - 0.1}$ $(\forall a \ge \frac{1}{2})$ (Qi & Luo, 2012), we get $\mathbb{E}\left[\sqrt{Z_k}\right] \le \sqrt{\frac{2(d_k/2 - 0.1)}{d_k}} = \sqrt{1 - \frac{0.2}{d_k}} \le 1 - \frac{0.1}{d_k}$. Therefore we have

$$\mathbb{E}\left[\sqrt{\prod_{k=i}^{j} Z_k}\right] \le \prod_{k=i}^{j}\left(1 - \frac{0.1}{d_k}\right) \le \left(1 - \frac{0.1}{O(L^{1-\gamma})}\right)^{j-i+1} \le \left(1 - \Omega(L^{\gamma-1})\right)^{\frac{L}{10}} = e^{-\Omega(L^\gamma)}.$$

Choose a sufficiently small constant $c' > 0$. By Markov inequality we have $\Pr\left[\sqrt{\prod_{k=i}^{j} Z_k} > e^{-c'L^\gamma}\right] \le e^{c'L^\gamma}\mathbb{E}\left[\sqrt{\prod_{k=i}^{j} Z_k}\right] \le e^{c'L^\gamma}e^{-\Omega(L^\gamma)} = e^{-\Omega(L^\gamma)}$. Therefore we have shown that for any fixed unit vector $v \in \mathbb{R}^{d_{i-1}}$, with probability at least $1 - e^{-\Omega(L^\gamma)}$ we have $\|A_{j:i}(0)v\| \le e^{-\Omega(L^\gamma)}$.

Next, we use this to bound $\|A_{j:i}(0)\|$ via an $\epsilon$-net argument. We partition the index set $[d_{i-1}]$ into $[d_{i-1}] = S_1 \cup S_2 \cup \cdots \cup S_q$ such that $|S_l| \le L^{\gamma/2}$ $(\forall l \in [q])$ and $q = O(\frac{d_{i-1}}{L^{\gamma/2}})$. For each $l \in [q]$, let $\mathcal{N}_l$ be a $\frac{1}{2}$-net for all the unit vectors in $\mathbb{R}^{d_{i-1}}$ with support in $S_l$. Note that we can choose $\mathcal{N}_l$ such that $|\mathcal{N}_l| = e^{O(|S_l|)} = e^{O(L^{\gamma/2})}$. Taking a union bound over $\cup_{l=1}^q \mathcal{N}_l$, we know that $\|A_{j:i}(0)v\| \le e^{-\Omega(L^\gamma)}\|v\|$ simultaneously for all $v \in \cup_{l=1}^q \mathcal{N}_l$ with probability at least $1 - \left(\sum_{l=1}^q |\mathcal{N}_l|\right)e^{-\Omega(L^\gamma)} \ge 1 - q \cdot e^{O(L^{\gamma/2})}e^{-\Omega(L^\gamma)} = 1 - e^{-\Omega(L^\gamma)}$.

Now, for any $u \in \mathbb{R}^{d_{i-1}}$, we write it as $u = \sum_{l=1}^q a_l u_l$ where $a_l$ is a scalar and $u_l$ is a unit vector supported on $S_l$. By the definition of $\frac{1}{2}$-net, for each $l \in [q]$ there exists $v_l \in \mathcal{N}_l$ such that $\|v_l - u_l\| \le \frac{1}{2}$. We know that $\|A_{j:i}(0)v_l\| \le e^{-\Omega(L^\gamma)}\|v_l\|$ for all $l \in [q]$. Let $v = \sum_{l=1}^q a_l v_l$. We have

$$\|A_{j:i}(0)v\| \le \sum_{l=1}^q |a_l| \cdot \|A_{j:i}(0)v_l\| \le \sum_{l=1}^q |a_l| \cdot e^{-\Omega(L^\gamma)}\|v_l\| \le e^{-\Omega(L^\gamma)}\sqrt{q \cdot \sum_{l=1}^q a_l^2 \|v_l\|^2}$$
$$= \sqrt{q}e^{-\Omega(L^\gamma)}\|v\| = e^{-\Omega(L^\gamma)}\|v\|.$$

Note that $\|u - v\| = \|\sum_{l=1}^q a_l(u_l - v_l)\| = \sqrt{\sum_{l=1}^q a_l^2 \|u_l - v_l\|^2} \le \sqrt{\frac{1}{4}\sum_{l=1}^q a_l^2} = \frac{1}{2}\|u\|$, which implies $\|v\| \le \frac{3}{2}\|u\|$. Therefore we have

$$\|A_{j:i}(0)u\| \le \|A_{j:i}(0)v\| + \|A_{j:i}(0)(u-v)\| \le e^{-\Omega(L^\gamma)}\|v\| + \|A_{j:i}(0)\| \cdot \|u - v\|$$
$$\le e^{-\Omega(L^\gamma)} \cdot \frac{3}{2}\|u\| + \|A_{j:i}(0)\| \cdot \frac{1}{2}\|u\| = e^{-\Omega(L^\gamma)}\|u\| + \|A_{j:i}(0)\| \cdot \frac{1}{2}\|u\|.$$

The above inequality is valid for any $u \in \mathbb{R}^{d_{i-1}}$. Thus we can take the unit vector $u$ that maximizes $\|A_{j:i}(0)u\|$. This gives us $\|A_{j:i}(0)\| \le e^{-\Omega(L^\gamma)} + \frac{1}{2}\|A_{j:i}(0)\|$, which implies $\|A_{j:i}(0)\| \le e^{-\Omega(L^\gamma)}$.

Finally, we take a union bound over all possible $(i, j)$. The failure probaility is at most $L^2 e^{-\Omega(L^\gamma)} = e^{-\Omega(L^\gamma)}$. $\qquad\square$

The following lemma shows that the properties in Lemmas 5.2 and 5.3 are still to some extent preserved after applying small perturbations on all the weight matrices.

**Lemma 5.4.** *Suppose that the initial weights satisfy $\|A_{j:i}(0)\| \leq O(L^3)$ for all $1 \leq i \leq j \leq L$, and $\|A_{j:i}(0)\| \leq e^{-c_1 L^\gamma}$ if $j - i \geq \frac{L}{10}$, where $c_1 > 0$ is a universal constant. Then for another set of matrices $A_1, \ldots, A_L$ satisfying $\|A_i - A_i(0)\| \leq e^{-0.6 c_1 L^\gamma}$ for all $i \in [L]$, we must have*

$$\|A_{j:i}\| \leq O(L^3), \quad \forall 1 \leq i \leq j \leq L,$$

$$\|A_{j:i}\| \leq O\left(e^{-c_1 L^\gamma}\right), \quad \forall 1 \leq i \leq j \leq L, j - i \geq \frac{L}{4}. \tag{12}$$

*Proof.* It suffices to show that the difference $A_{j:i} - A_{j:i}(0)$ is tiny. Let $\Delta_i = A_i - A_i(0)$. We have $A_{j:i} = (A_j(0) + \Delta_j) \cdots (A_{i+1}(0) + \Delta_{i+1})(A_i(0) + \Delta_i)$. Expanding this product, except for the one term corresponding to $A_{j:i}(0)$, every other term has the form $A_{j:(k_s+1)}(0) \cdot \Delta_{k_s} \cdot A_{(k_s-1):(k_{s-1}+1)}(0) \cdot \Delta_{k_{s-1}} \cdots \Delta_{k_1} \cdot A_{(k_1-1):i}(0)$, where $i \leq k_1 < \cdots < k_s \leq j$. By assumption, each $\Delta_k$ has spectral norm $e^{-0.6 c_1 L^\gamma}$, and each $A_{j':i'}(0)$ has spectral norm $O(L^3)$, so we have $\left\|A_{j:(k_s+1)}(0) \cdot \Delta_{k_s} \cdot A_{(k_s-1):(k_{s-1}+1)}(0) \cdot \Delta_{k_{s-1}} \cdots \Delta_{k_1} \cdot A_{(k_1-1):i}(0)\right\| \leq \left(e^{-0.6 c_1 L^\gamma}\right)^s \left(O(L^3)\right)^{s+1}$. Therefore we have

$$\|A_{j:i} - A_{j:i}(0)\| \leq \sum_{s=1}^{j-i+1} \binom{j-i+1}{s} \left(e^{-0.6 c_1 L^\gamma}\right)^s \left(O(L^3)\right)^{s+1}$$

$$\leq \sum_{s=1}^{j-i+1} L^s \left(e^{-0.6 c_1 L^\gamma}\right)^s \left(O(L^3)\right)^{s+1} \leq O(L^3) \sum_{s=1}^{\infty} \left(O(L^4) e^{-0.6 c_1 L^\gamma}\right)^s \leq O(L^3) \sum_{s=1}^{\infty} (1/2)^s = O(L^3),$$

which implies $\|A_{j:i}\| \leq O(L^3)$ for all $1 \leq i \leq j \leq L$.

The proof of the second part of the lemma is postponed to Appendix B.2. □

As a consequence of Lemma 5.4, we can control the objective value and the gradient at any point sufficiently close to the random initialization.

**Lemma 5.5.** *For a set of weight matrices $W_1, \ldots, W_L$ with $A_i = W_i/(\sqrt{d_i}\sigma_i)$ that satisfy (12), the objective and the gradient satisfy*

$$0.4 \|Y\|_F^2 < \ell(W_1, \ldots, W_L) < 0.6 \|Y\|_F^2,$$

$$\|\nabla_{W_i} \ell(W_1, \ldots, W_L)\| \leq (\sqrt{d_i}\sigma_i)^{-1} e^{-0.9 c_1 L^\gamma}, \quad \forall i \in [L].$$

The proof of Lemma 5.5 is given in Appendix B.3.

Finally, we can finish the proof of Theorem 5.1 using the above lemmas.

*Proof of Theorem 5.1.* From Lemmas 5.2 and 5.3, we know that with probability at least 0.9, we have (i) $\|A_{j:i}(0)\| \leq O(L^3)$ for all $1 \leq i \leq j \leq L$, and (ii) $\|A_{j:i}(0)\| \leq e^{-c_1 L^\gamma}$ if $(i, j)$ further satisfies $j - i \geq \frac{L}{10}$. Here $c_1 > 0$ is a universal constant. From now on we are conditioned on these properties being satisfied. We suppose that the learning rate $\eta$ is at most $e^{0.2 c_1 L^\gamma}$.

We say that a set of weight matrices $W_1, \ldots, W_L$ are in the "initial neighborhood" if $\|A_i - A_i(0)\| \leq e^{-0.6 c_1 L^\gamma}$ for all $i \in [L]$. From Lemmas 5.4 and 5.5 we know that in the "initial neighborhood" the objective value is always between $0.4 \|Y\|_F^2$ and $0.6 \|Y\|_F^2$. Therefore we have to escape the "initial neighborhood" in order to get the objective value out of this interval.

Now we calculate how many iterations are necessary to escape the "initial neighborhood." According to Lemma 5.5, inside the "initial neighborhood" each $W_i$ can move at most $\eta(\sqrt{d_i}\sigma_i)^{-1} e^{-0.9 c_1 L^\gamma}$ in one iteration by definition of the gradient descent algorithm. In order to leave the "initial neighborhood," some $W_i$ must satisfy $\|W_i - W_i(0)\| = \sqrt{d_i}\sigma_i \|A_i - A_i(0)\| > \sqrt{d_i}\sigma_i e^{-0.6 c_1 L^\gamma}$. In order to move this amount, the number of iterations has to be at least

$$\frac{\sqrt{d_i}\sigma_i e^{-0.6 c_1 L^\gamma}}{\eta(\sqrt{d_i}\sigma_i)^{-1} e^{-0.9 c_1 L^\gamma}} = \frac{d_i \sigma_i^2 e^{0.3 c_1 L^\gamma}}{\eta} \geq \frac{1}{L^{O(1)}} \cdot \frac{e^{0.3 c_1 L^\gamma}}{e^{0.2 c_1 L^\gamma}} \geq e^{\Omega(L^\gamma)}.$$

This finishes the proof. □

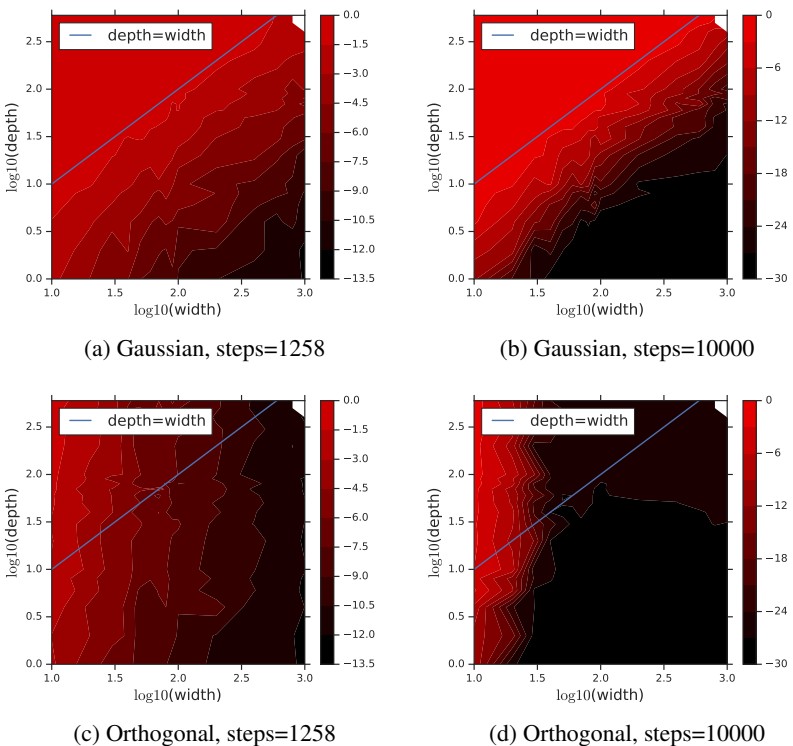

Figure 1: $\log \frac{\ell(t)}{\ell(0)}$ at $t = 1258$ and $t = 10000$, for different depth-width configurations and different initialization schemes. Darker color means smaller loss.

## 6 EXPERIMENTS

In this section, we provide empirical evidence to support the results in Sections 4 and 5. To study how depth and width affect convergence speed of gradient descent under orthogonal and Gaussian initialization schemes, we train a family of linear networks with their widths ranging from 10 to 1000 and depths from 1 to 700, on a fixed synthetic dataset $(X, Y)$.[4] Each network is trained using gradient descent staring from both Gaussian and orthogonal initializations. In Figure 1, We lay out the logarithm of the relative training loss $\frac{\ell(t)}{\ell(0)}$, using heap-maps, at steps $t = 1258$ and $t = 10000$. In each heat-map, each point represents the relative training loss of one experiment; the darker the color, the smaller the loss. Figure 1 clearly demonstrates a sharp transition from untrainable to trainable (i.e., from red to black) when we increase the width of the network:

- for Gaussian initialization, this transition occurs across a contour characterized by a linear relation between width and depth;

- for orthogonal initialization, the transition occurs at a width that is approximately independent of the depth.

These observations excellently verify our theory developed in Sections 4 and 5.

To have a closer look into the training dynamics, we also plot "relative loss v.s. training time" for a variety of depth-width configurations. See Figure 2. There again we can clearly see that orthogonal initialization enables fast training at small width (independent of depth), and that the required width for Gaussian initialization depends on depth.

---

[4]We choose $X \in \mathbb{R}^{1024 \times 16}$ and $W^* \in \mathbb{R}^{10 \times 1024}$, and set $Y = W^* X$. Entries in $X$ and $W^*$ are drawn i.i.d. from $\mathcal{N}(0, 1)$.

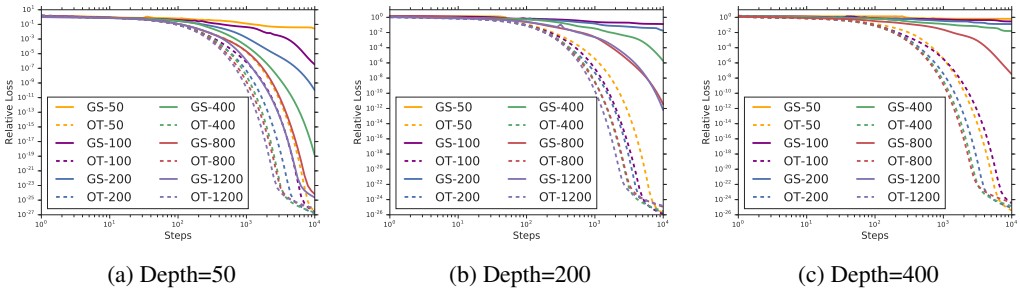

| (a) Depth=50 | (b) Depth=200 | (c) Depth=400 |

Figure 2: Relative loss v.s. training time. For each plot, we vary width from 50 (yellow) to 1200 (purple). Solid and dashed lines represent Gaussian (GS) and orthogonal (OT) initializations.

## 7    CONCLUSION

In this work, we studied the effect of the initialization parameter values of deep linear neural networks on the convergence time of gradient descent. We found that when the initial weights are iid Gaussian, the convergence time grows exponentially in the depth unless the width is at least as large as the depth. In contrast, when the initial weight matrices are drawn from the orthogonal group, the width needed to guarantee efficient convergence is in fact independent of the depth. These results establish for the first time a concrete proof that orthogonal initialization is superior to Gaussian initialization in terms of convergence time.

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

# A  PROOFS FOR SECTION 4

## A.1  PROOF OF LEMMA 4.2

*Proof of Lemma 4.2.* We only need to prove (9). We first upper bound the magnitude of the network's initial output on any given input $x \in \mathbb{R}^{d_x}$. Let $z = \frac{1}{\sqrt{m^{L-1}}} W_{L-1:1}(0) \cdot x \in \mathbb{R}^m$. Then we have $\|z\| = \|x\|$, and $f(x; W_1(0), \ldots, W_L(0)) = \frac{1}{\sqrt{d_y}} W_L(0) \cdot z = \sqrt{\frac{m}{d_y}} \cdot \frac{1}{\sqrt{m}} W_L(0) \cdot z$. Note that $\frac{1}{\sqrt{m}} W_L(0) \cdot z$ is the (signed) projection of $z$ onto a random subspace in $\mathbb{R}^m$ of dimension $d_y$. Therefore $\left\| \frac{1}{\sqrt{m}} W_L(0) \cdot z \right\|^2 / \|z\|^2$ has the same distribution as $\frac{g_1^2 + \cdots + g_{d_y}^2}{g_1^2 + \cdots + g_m^2}$, where $g_1, \ldots, g_m$ are i.i.d. samples from $\mathcal{N}(0, 1)$. By the standard tail bounds for chi-squared distributions we have

$$\Pr\left[ g_1^2 + \cdots + g_{d_y}^2 \le d_y + 2\sqrt{d_y \log(1/\delta')} + 2\log(1/\delta') \right] \ge 1 - \delta',$$

$$\Pr\left[ g_1^2 + \cdots + g_m^2 \ge m - 2\sqrt{m \log(1/\delta')} \right] \ge 1 - \delta'.$$

Let $\delta' = \frac{\delta}{2r}$. Note that $m > C \cdot \log(r/\delta)$. We know that with probability at least $1 - \frac{\delta}{r}$ we have

$$\left\| \frac{1}{\sqrt{m}} W_L(0) \cdot z \right\|^2 / \|z\|^2 \le \frac{d_y + 2\sqrt{d_y \log(2r/\delta)} + 2\log(2r/\delta)}{m - 2\sqrt{m \log(2r/\delta)}} = \frac{O(d_y + \log(r/\delta))}{\Omega(m)},$$

which implies

$$\|f(x; W_1(0), \ldots, W_L(0))\|^2 = \frac{m}{d_y} \left\| \frac{1}{\sqrt{m}} W_L(0) \cdot z \right\|^2 = \frac{m}{d_y} \cdot O\left( \frac{d_y + \log(r/\delta)}{m} \right) \|z\|^2$$

$$= O\left( 1 + \frac{\log(r/\delta)}{d_y} \right) \|x\|^2. \tag{13}$$

Finally, taking a union bound, we know that with probability at least $1 - \delta$, the inequality (13) holds for every $x \in \{x_1, \ldots, x_r\}$, which implies

$$\ell(0) = \frac{1}{2} \sum_{k=1}^r \|f(x_k; W_1(0), \ldots, W_L(0)) - y_k\|^2 \le \sum_{k=1}^r \left( \|f(x_k; W_1(0), \ldots, W_L(0))\|^2 + \|y_k\|^2 \right)$$

$$\le O\left( 1 + \frac{\log(r/\delta)}{d_y} \right) \sum_{k=1}^r \|x_k\|^2 + \sum_{k=1}^r \|y_k\|^2 = O\left( 1 + \frac{\log(r/\delta)}{d_y} \right) \|X\|_F^2 + \|Y\|_F^2$$

$$\le O\left( 1 + \frac{\log(r/\delta)}{d_y} + \|W^*\|^2 \right) \|X\|_F^2. \qquad \square$$

## A.2  PROOF OF CLAIM 4.3

*Proof of Claim 4.3.* Let $\gamma = \frac{1}{2} L \sigma_{\min}^2(X)/d_y$. From $\mathcal{A}(0), \ldots, \mathcal{A}(t)$ we have $\ell(s) \le (1 - \eta\gamma)^s B$ for all $0 \le s \le t$. The gradient of the objective function (2) is $\frac{\partial \ell}{\partial W_i} = \alpha W_{L:i+1}^\top (U - Y) (W_{i-1:1} X)^\top$. Thus we can bound the gradient norm as follows for all $0 \le s \le t$ and all $i \in [L]$:

$$\left\| \frac{\partial \ell}{\partial W_i}(s) \right\|_F \le \alpha \|W_{L:i+1}(s)\| \|U(s) - Y\|_F \|W_{i-1:1}(s)\| \|X\|$$

$$\le \frac{1}{\sqrt{m^{L-1} d_y}} \cdot 1.1 m^{\frac{L-i}{2}} \cdot \sqrt{2\ell(s)} \cdot 1.1 m^{\frac{i-1}{2}} \|X\| \le \frac{2\sqrt{(1 - \eta\gamma)^s B}}{\sqrt{d_y}} \|X\|, \tag{14}$$

where we have used $\mathcal{B}(s)$. Then for all $i \in [L]$ we have:

$$\|W_i(t+1) - W_i(0)\|_F \le \sum_{s=0}^t \|W_i(s+1) - W_i(s)\|_F = \sum_{s=0}^t \left\| \eta \frac{\partial \ell}{\partial W_i}(s) \right\|_F$$

$$\leq \eta \sum_{s=0}^{t} \frac{2\sqrt{(1-\eta\gamma)^s B}}{\sqrt{d_y}} \|X\| \leq \frac{2\eta\sqrt{B}}{\sqrt{d_y}} \|X\| \sum_{s=0}^{t-1} (1-\eta\gamma/2)^s \leq \frac{2\eta\sqrt{B}}{\sqrt{d_y}} \|X\| \cdot \frac{2}{\eta\gamma}$$

$$= \frac{8\sqrt{Bd_y}\,\|X\|}{L\sigma_{\min}^2(X)}.$$

This proves $\mathcal{C}(t+1)$. $\qquad\square$

## A.3  Proof of Claim 4.4

*Proof of Claim 4.4.* Let $R = \frac{8\sqrt{Bd_y}\|X\|}{L\sigma_{\min}^2(X)}$ and $\Delta_i = W_i(t) - W_i(0)$ $(i \in [L])$. Then $\mathcal{C}(t)$ means $\|\Delta_i\|_F \leq R\,(\forall i \in [L])$.

For $1 \leq i \leq j \leq L$, we have

$$W_{j:i}(t) = (W_j(0) + \Delta_j)\cdots(W_i(0) + \Delta_i).$$

Expanding this product, each term except $W_{j:i}(0)$ has the form:

$$W_{j:(k_s+1)}(0) \cdot \Delta_{k_s} \cdot W_{(k_s-1):(k_{s-1}+1)}(0) \cdot \Delta_{k_{s-1}} \cdots \Delta_{k_1} \cdot W_{(k_1-1):i}(0), \qquad (15)$$

where $i \leq k_1 < \cdots < k_s \leq j$ are locations where terms like $\Delta_{k_l}$ are taken out. Note that every factor in (15) of the form $W_{j':i'}(0)$ satisfies $\|W_{j':i'}(0)\| = m^{\frac{j'-i'+1}{2}}$ according to (8). Thus, we can bound the sum of all terms of the form (15) as

$$\|W_{j:i}(t) - W_{j:i}(0)\| \leq \sum_{s=1}^{j-i+1} \binom{j-i+1}{s} R^s m^{\frac{j-i+1-s}{2}} = (\sqrt{m} + R)^{j-i+1} - (\sqrt{m})^{j-i+1}$$

$$= (\sqrt{m})^{j-i+1}\left((1 + R/\sqrt{m})^{j-i+1} - 1\right) \leq (\sqrt{m})^{j-i+1}\left((1 + R/\sqrt{m})^L - 1\right) \leq 0.1(\sqrt{m})^{j-i+1}.$$

Here the last step uses $m > C(LR)^2$ which is implied by (5). Combined with (8), this proves $\mathcal{B}(t)$. $\qquad\square$

## A.4  Proof of Claim 4.5

*Proof of Claim 4.5.* Recall that we have the dynamics (6) for $U(t)$. In order to establish convergence from (6) we need to prove upper and lower bounds on the eigenvalues of $P(t)$, as well as show that the high-order term $E(t)$ is small. We will prove these using $\mathcal{B}(t)$.

Using the definition (7) and property $\mathcal{B}(t)$, we have

$$\lambda_{\max}(P(t)) \leq \alpha^2 \sum_{i=1}^{L} \lambda_{\max}\left((W_{i-1:1}(t)X)^\top (W_{i-1:1}(t)X)\right) \cdot \lambda_{\max}\left(W_{L:i+1}(t)W_{L:i+1}^\top(t)\right)$$

$$\leq \frac{1}{m^{L-1}d_y} \sum_{i=1}^{L} \left(1.1m^{\frac{i-1}{2}}\sigma_{\max}(X)\right)^2 \left(1.1m^{\frac{L-i}{2}}\right)^2 \leq 2L\sigma_{\max}^2(X)/d_y,$$

$$\lambda_{\min}(P(t)) \geq \alpha^2 \sum_{i=1}^{L} \lambda_{\min}\left((W_{i-1:1}(t)X)^\top (W_{i-1:1}(t)X)\right) \cdot \lambda_{\min}\left(W_{L:i+1}(t)W_{L:i+1}^\top(t)\right)$$

$$\geq \frac{1}{m^{L-1}d_y} \sum_{i=1}^{L} \left(0.9m^{\frac{i-1}{2}}\sigma_{\min}(X)\right)^2 \left(0.9m^{\frac{L-i}{2}}\right)^2 \geq \frac{3}{5}L\sigma_{\min}^2(X)/d_y.$$

In the lower bound above, we make use of the following relation on dimensions: $m \geq d_x \geq r$, which enables the inequality $\lambda_{\min}\left((W_{i-1:1}(t)X)^\top (W_{i-1:1}(t)X)\right) = \sigma_{\min}^2(W_{i-1:1}(t)X) \geq \sigma_{\min}^2(W_{i-1:1}(t)) \cdot \sigma_{\min}^2(X)$.

Next, we will prove the following bound on the high-order term $E(t)$:

$$\frac{1}{\sqrt{m^{L-1}d_y}} \|E(t)X\|_F \leq \frac{1}{6}\eta\lambda_{\min}(P_t)\|U(t) - Y\|_F.$$

Recall that $E(t)$ is the sum of all high-order terms in the product

$$W_{L:1}(t+1) = \prod_i \left( W_i(t) - \eta \frac{\partial \ell}{\partial W_i}(t) \right).$$

Same as (14), we have $\left\| \frac{\partial \ell}{\partial W_i}(t) \right\|_F \leq \frac{2\sqrt{\ell(t)}\|X\|}{\sqrt{d_y}}$ ($\forall i \in [L]$). Then we have

$$\frac{1}{\sqrt{m^{L-1}d_y}} \|E(t)X\|_F$$

$$\leq \frac{1}{\sqrt{m^{L-1}d_y}} \sum_{s=2}^{L} \binom{L}{s} \left( \eta \cdot \frac{2\sqrt{\ell(t)}\|X\|}{\sqrt{d_y}} \right)^s m^{\frac{L-s}{2}} \|X\|$$

$$\leq \sqrt{\frac{m}{d_y}} \|X\| \sum_{s=2}^{L} L^s \left( \eta \cdot \frac{2\sqrt{\ell(t)}\|X\|}{\sqrt{d_y}} \right)^s m^{-\frac{s}{2}}$$

$$= \sqrt{\frac{m}{d_y}} \|X\| \sum_{s=2}^{L} \left( \frac{2\eta L \sqrt{\ell(t)}\|X\|}{\sqrt{md_y}} \right)^s$$

From $\eta \leq \frac{d_y}{2L\|X\|^2}$, we have $\frac{2\eta L\sqrt{\ell(t)}\|X\|}{\sqrt{md_y}} \leq \frac{\sqrt{d_y \cdot \ell(t)}}{\sqrt{m}\|X\|}$. Note that $m > C \cdot \frac{d_y B}{\|X\|^2} \geq C \cdot \frac{d_y \ell(t)}{\|X\|^2}$. Thus we have

$$\frac{1}{\sqrt{m^{L-1}d_y}} \|E(t)X\|_F \leq \sqrt{\frac{m}{d_y}} \|X\| \left( \frac{2\eta L\sqrt{\ell(t)}\|X\|}{\sqrt{md_y}} \right)^2 \sum_{s=2}^{L-2} 0.5^{s-2}$$

$$\leq 2\sqrt{\frac{m}{d_y}} \|X\| \left( \frac{2\eta L\sqrt{\ell(t)}\|X\|}{\sqrt{md_y}} \right)^2 \leq 2\sqrt{\frac{m}{d_y}} \|X\| \cdot \frac{2\eta L\sqrt{\ell(t)}\|X\|}{\sqrt{md_y}} \cdot \frac{\sqrt{d_y \cdot \ell(t)}}{\sqrt{m}\|X\|}$$

$$= \frac{4\eta L\|X\| \cdot \ell(t)}{\sqrt{md_y}}.$$

It suffices to show that the above bound is at most $\frac{1}{6}\eta\lambda_{\min}(P_t)\|U(t)-Y\|_F = \frac{1}{6}\eta\lambda_{\min}(P_t)\sqrt{2\ell(t)}$. Since $\lambda_{\min}(P_t) \geq \frac{3}{5}L\sigma_{\min}^2(X)/d_y$, it suffices to have

$$\frac{4\eta L\|X\| \cdot \ell(t)}{\sqrt{md_y}} \leq \frac{1}{6}\eta \cdot \frac{3L\sigma_{\min}^2(X)\sqrt{2\ell(t)}}{5d_y},$$

which is true since $m > C \cdot \frac{d_y B\|X\|^2}{\sigma_{\min}^4(X)} \geq C \cdot \frac{d_y \ell(t)\|X\|^2}{\sigma_{\min}^4(X)}$.

Finally, from (6) and $\eta \leq \frac{d_y}{2L\|X\|^2} \leq \frac{1}{\lambda_{\max}(P_t)}$ we have

$$\|U(t+1) - Y\|_F = \|\mathrm{vec}\,(U(t+1) - Y)\|$$

$$= \left\| (I - \eta P(t)) \cdot \mathrm{vec}\,(U(t) - Y) + \frac{1}{\sqrt{m^{L-1}d_y}} \mathrm{vec}\,(E(t)X) \right\|$$

$$\leq (1 - \eta\lambda_{\min}(P(t))) \|\mathrm{vec}\,(U(t) - Y)\| + \frac{1}{\sqrt{m^{L-1}d_y}} \|E(t)X\|_F$$

$$\leq (1 - \eta\lambda_{\min}(P(t))) \|U(t) - Y\|_F + \frac{1}{6}\eta\lambda_{\min}(P_t) \|U(t) - Y\|_F$$

$$= \left( 1 - \frac{5}{6}\eta\lambda_{\min}(P(t)) \right) \|U(t) - Y\|_F$$

$$\leq \left( 1 - \frac{1}{2}\eta L\sigma_{\min}^2(X)/d_y \right) \|U(t) - Y\|_F.$$

Therefore $\ell(t+1) \leq \left( 1 - \frac{1}{2}\eta L\sigma_{\min}^2(X)/d_y \right)^2 \ell(t) \leq \left( 1 - \frac{1}{2}\eta L\sigma_{\min}^2(X)/d_y \right) \ell(t)$. Combined with $\mathcal{A}(t)$, this proves $\mathcal{A}(t+1)$. $\qquad \square$

# B  PROOFS FOR SECTION 5

## B.1  PROOF OF LEMMA 5.2

*Proof of Lemma 5.2.* Notice that for any $1 \le i \le j \le L$ we have $\mathbb{E}\left[\|A_{j:i}(0)\|_F^2\right] = d_{i-1}$. Then by Markov inequality we have $\Pr\left[\|A_{j:i}(0)\|_F^2 \ge \frac{d_{i-1}}{\delta/L^2}\right] \le \delta/L^2$. Taking a union bound, we know that with probability at least $1 - \delta$, for all $1 \le i \le j \le L$ simultaneously we have $\|A_{j:i}(0)\| \le \|A_{j:i}(0)\|_F \le \frac{d_{i-1}}{\delta/L^2} \le O(L^3/\delta)$ (note that $d_{i-1} \le O(L^{1-\gamma}) = O(L)$). $\square$

## B.2  PROOF OF LEMMA 5.4

*Proof of Lemma 5.4 (continued).* For the second part of the lemma ($j - i \ge \frac{L}{4}$), we need to bound the terms of the form $A_{j:k+1}(0) \cdot \Delta_k \cdot A_{k-1:i}(0)$ more carefully. In fact, if $j - i \ge \frac{L}{4}$, then $\max\{j - k - 1, k - 1 - i\} \ge \frac{L}{10}$, which by assumption means either $A_{j:k+1}(0)$ or $A_{k-1:i}(0)$ has spectral norm bounded by $e^{-c_1 L^\gamma}$. This implies $\|A_{j:k+1}(0) \cdot \Delta_k \cdot A_{k-1:i}(0)\| \le e^{-c_1 L^\gamma} e^{-0.6 c_1 L^\gamma} \cdot O(L^3) = e^{-1.6 c_1 L^\gamma} \cdot O(L^3)$. Therefore we have

$$\|A_{j:i} - A_{j:i}(0)\| \le (j - i + 1) e^{-1.6 c_1 L^\gamma} \cdot O(L^3) + \sum_{s=2}^{j-i+1} \binom{j-i+1}{s} \left(e^{-0.6 c_1 L^\gamma}\right)^s \left(O(L^3)\right)^{s+1}$$

$$\le e^{-c_1 L^\gamma} + \sum_{s=2}^{\infty} L^s \left(e^{-0.6 c_1 L^\gamma}\right)^s \left(O(L^3)\right)^{s+1} \le e^{-c_1 L^\gamma} + \sum_{s=2}^{\infty} \left(e^{-0.5 c_1 L^\gamma}\right)^s = O\left(e^{-c_1 L^\gamma}\right).$$

This implies $\|A_{j:i}\| \le O\left(e^{-c_1 L^\gamma}\right)$. $\square$

## B.3  PROOF OF LEMMA 5.5

*Proof of Lemma 5.5.* We can bound the network's output as

$$\|\alpha W_{L:1}(0) X\|_F = \|\beta A_{L:1}(0) X\|_F \le L^{O(1)} \cdot e^{-\Omega(L^\gamma)} \|X\|_F = e^{-\Omega(L^\gamma)}.$$

Thus the objective value $\ell(W_1, \ldots, W_L) = \frac{1}{2} \|\alpha W_{L:1}(0) X - Y\|_F^2$ must be extremely close to $\frac{1}{2} \|Y\|_F^2$ for large $L$, so $0.4 \|Y\|_F^2 < \ell(W_1, \ldots, W_L) < 0.6 \|Y\|_F^2$.

As for the gradient, for any $i \in [L]$ we have

$$\|\nabla_{W_i} \ell(W_1, \ldots, W_L)\| = \|\alpha W_{L:i+1}^\top (\alpha W_{L:1} X - Y) X^\top W_{i-1:1}^\top\|$$

$$= \left\|\beta/(\sqrt{d_i}\sigma_i) \cdot A_{L:i+1}^\top (\alpha W_{L:1} X - Y) X^\top A_{i-1:1}^\top\right\| \le \frac{L^{O(1)}}{\sqrt{d_i}\sigma_i} \|A_{L:i+1}\| \cdot O(1) \cdot \|A_{i-1:1}\|.$$

Using (12), and noting that either $L - i - 1$ or $i - 1$ is greater than $\frac{L}{4}$, we have

$$\|\nabla_{W_i} \ell(W_1, \ldots, W_L)\| \le \sigma_i^{-1} L^{O(1)} \cdot O\left(e^{-c_1 L^\gamma}\right) \cdot O(L^3) \le (\sqrt{d_i}\sigma_i)^{-1} e^{-0.9 c_1 L^\gamma}. \qquad \square$$

