# OpenReview forum: "Provable Benefit of Orthogonal Initialization in Optimizing Deep Linear Networks"
_ICLR.cc/2020/Conference — Accept (Poster)_

### Official Review · AnonReviewer2 · 2019-10-23
**Official Blind Review #2**

**Rating:** 8

**Review:**

This paper rigorously proves that if a deep linear network is initialized with random orthogonal weights and trained with gradient descent, its width required for convergence  does not depend on its depth. To compare, when weights in deep linear networks are initialized with Gaussian initialization, the minimal width required for convergence will depend on the depth of the network. This proof explains why orthogonal weight initialization can help to train networks efficiently, especially for those very deep ones.

The theoretical contribution of this paper is very important. Orthogonal initialization is found to be useful in deep network training. Although the theory in this paper is developed for linear networks, it still has important guidance meaning in practices in more areas of deep learning. The derivations are correct to my best knowledge. And the paper is well-written and easy to read.

Minor points:
- typo in the last equation in (4)

=======================
Update: Despite the similarity with a previous paper, I still think the theoretical results and empirical observations important and thus I will keep my score.

**Experience Assessment:**

I have published one or two papers in this area.

**Review Assessment: Checking Correctness Of Derivations And Theory:**

I assessed the sensibility of the derivations and theory.

**Review Assessment: Checking Correctness Of Experiments:**

N/A

**Review Assessment: Thoroughness In Paper Reading:**

I read the paper at least twice and used my best judgement in assessing the paper.

---

> ### Author Response · Authors · 2019-11-15
> **Response to Reviewer #2**
>
> Thank you for your encouraging review!

---

### Official Review · AnonReviewer1 · 2019-10-23
**Official Blind Review #1**

**Rating:** 3

**Review:**

This paper studies the convergence of deep linear networks under orthogonal initialization. Most of the problem setup, analysis techniques and proof roadmap are adapted from Du & Hu (2019). The difference is the orthogonal initialization compared with the Gaussian initialization in Du & Hu (2019). Since the product of orthogonal weight matrices is identity, the new initialization can remove the dependency of the number of nodes $m$ on the depth $L$ of the network. The authors also proved a lower bound of the loss function trained by Gaussian initialization within certain iterations. This justifies the disadvantage of Gaussian initialization.

My biggest concern is that this paper seems to be very similar to Du & Hu (2019) in many places. The whole Sections 3 & 4 are almost the same as Sections 3,4,5 & 7 in their paper. The contribution in this paper is too incremental given previous work.

What is the advantage of using orthogonal matrices in (4) compared with just using identity matrices as initialization? Can we prove the same result with just identity initialization? In this case, what would we lose by restricting us to this special case?

In the proof of Lemma 4.2, it seems that only the randomness of the last layer $W_L(0)$ is used. Why do we need all the layers to be uniformly sampled?

It should be explained in more details that $W_L(0)$ is drawn from a uniform distribution over orthogonal matrices in $d_y\times d_y$ space. Then $1/\sqrt{m} W_L(0)\cdot z /\|z\|^2$ is not distributed on the whole space of $d_y$-sphere. The argument in the proof of Lemma 4.2 thus needs more justification.

It would be interesting to see an empirical comparison of the proposed initialization and the Gaussian initialization. Due to the lower bound proved in this paper, the experiments are expected to show distinct difference between these two.


**Experience Assessment:**

I have read many papers in this area.

**Review Assessment: Checking Correctness Of Derivations And Theory:**

I assessed the sensibility of the derivations and theory.

**Review Assessment: Checking Correctness Of Experiments:**

N/A

**Review Assessment: Thoroughness In Paper Reading:**

I read the paper at least twice and used my best judgement in assessing the paper.

---

> ### Author Response · Authors · 2019-11-15
> **Response to Reviewer #1**
>
> Thank you for your valuable comments. Please see our response to each individual question below.
>
>
> — Similarity to Du and Hu (2019) —
> While the proof technique for our result on orthogonal initialization (Thm 4.1) is similar to that of Du and Hu (2019), we’d like to stress that the contribution of this paper is much more than the specific proof of this theorem.
>
> First of all, we believe that the main contribution of this paper should be the results themselves rather than the specific proof techniques. We establish rigorously that orthogonal initialization can drastically speed up optimization compared with Gaussian initialization in deep linear networks, which was empirically observed in a lot of previous work without a formal theoretical justification except for initialization.
>
> Furthermore, our negative result on Gaussian initialization (Thm 5.1) is an important piece of the paper (which takes up at least half of the technical part of the paper). This result is novel - there was no such result in Du and Hu (2019), and its proof is not adapted from any previous work (see our response to Reviewer 3).
>
>
> — Can use identity matrices in intermediate layers & only the randomness of the last layer is used —
> You are correct. Thanks for pointing out! It is true that we can just use identity initialization in intermediate layers. We think this is due to the specialness of linear networks. When generalizing this result to deep non-linear networks (on-going work), the randomness in all layers becomes important. Such randomness was also used in previous work that studied orthogonal random initializations, e.g. Pennington et al. (2017, 2018).
>
> [Pennington et al. (2017)] Resurrecting the sigmoid in deep learning through dynamical isometry: theory and practice.
> [Pennington et al. (2018)] The emergence of spectral universality in deep networks.
>
>
> — Distribution of $W_L(0)$ —
> $W_L(0)$ has size $d_y \times m (d_y \le m)$ and is drawn from the uniform measure over all row-orthogonal matrices satisfying $W_L(0)W_L(0)^\top = mI_{d_y}$. Then $\frac{1}{\sqrt m} W_L(0)$ has unit-length orthogonal rows. When $\frac{1}{\sqrt m} W_L(0)$ is multiplied by $z \in \mathbb{R}^m$, it’s effectively projecting $z$ onto $d_y$ random directions (corresponding to rows of $\frac{1}{\sqrt m} W_L(0)$), and as a result we can apply concentration bounds on the norm of this product. Thank you for raising the confusion, and we will explain this more clearly in the paper.
>
>
> — Experiment —
> Thank you for the suggestion! We have added a short experiment section (Section 6) to the paper. We train a family of deep linear networks with their widths ranging in [10, 1000] and depths ranging in [1, 700]. Each network is trained on the same dataset using gradient descent starting from both Gaussian and orthogonal initialization, and we produce heat-maps whose colors exhibit the losses after 10,000 steps for all configurations.
>
> The heat-maps clearly demonstrate a sharp transition from untrainable to trainable when we increase the width of the network. For Gaussian initialization, this transition occurs across a contour characterized by a linear relation between width and depth; for orthogonal initialization, the transition occurs at a width that is approximately independent of the depth. These observations match our theory excellently. Please see Section 6 for details.

---

### Official Review · AnonReviewer3 · 2019-10-24
**Official Blind Review #3**

**Rating:** 6

**Review:**

This paper studies the role of initialization for training deep linear neural networks. The authors specifically consider the orthogonal initialization, and prove that with the orthogonal initialization proposed in equation (4), the gradient descent can achieve zero training error in a linear convergence rate. The improvement of the orthogonal initialization lies at the dependence of the layer width $m$, which is independent of the network depth $L$.

The problem considered in this work is very interesting since there are lots of empirical studies show that good initialization can benefit the training of deep neural networks. However, my main concern about this work is its novelty, especially for the proof techniques used in the current paper. It seems that most of the proofs are similar to the previous work Du & Hu (2019), and the main reason that it can remove the dependence of $L$ seems to be Lemma 4.2, which can be derived using the orthogonal property of the initialization. In this sense, there is not too much contribution for the current paper given the previous work Du & Hu (2019). Are there any other significant changes need to be made in the proofs to get the main results? If the authors can provide the convergence guarantees of the stochastic gradient descent, the contributions would be strong.

For the proof of Theorem 5.1, is it a straightforward extension of the proofs in Shamir (2018)? What is the main challenge when prove the general $d$ case?


Minor comments:
For the last equation in (4), why you need $W_L(0)W_L(0)^\top=mI_{d_y}$ instead of $W_L(0)^\top W_L(0)=d_yI$ as you used in the later proofs?

There is no experiment to verify the theory

Update:
I thank the authors for their response, I would like to keep my score.

**Experience Assessment:**

I have published one or two papers in this area.

**Review Assessment: Checking Correctness Of Derivations And Theory:**

I assessed the sensibility of the derivations and theory.

**Review Assessment: Checking Correctness Of Experiments:**

N/A

**Review Assessment: Thoroughness In Paper Reading:**

I read the paper thoroughly.

---

> ### Author Response · Authors · 2019-11-15
> **Response to Reviewer #3**
>
> Thank you for your valuable comments and for appreciating our work. Please see our response to each individual question below.
>
>
> — Similarity to Du and Hu (2019) —
> While the proof technique for our result on orthogonal initialization (Thm 4.1) is similar to that of Du and Hu (2019), we’d like to stress that the contribution of this paper is much more than the specific proof of this theorem.
>
> First of all, we believe that the main contribution of this paper should be the results themselves rather than the specific proof techniques. We establish rigorously that orthogonal initialization can drastically speed up optimization compared with Gaussian initialization in deep linear networks, which was empirically observed in a lot of previous work without a formal theoretical justification except for initialization.
>
> Furthermore, our negative result on Gaussian initialization (Thm 5.1) is an important piece of the paper (which takes up at least half of the technical part of the paper). This result is novel - there was no such result in Du and Hu (2019), and its proof is not adapted from any previous work  (see below).
>
>
> — Is the proof of Thm 5.1 a straightforward extension of Shamir (2018)? —
> The proof is not a straightforward extension of Shamir (2018). Shamir’s proof relies on several special properties of 1-dim linear networks, such as a gradient norm bound for weights near initialization (Lemma 3 in his paper). Due to such specialness, his proof cannot be extended to any dimension greater than 1, as explicitly mentioned in his paper. Our result not only deals with multiple dimensions, but also reveals a nearly tight relation between width and depth for trainability (see paragraph after Thm 5.1). Key to our analysis is a careful control on the spectral norm of $A_{j:i}$ at initialization (Lemma 5.3) and after perturbations during training (Lemma 5.4). An important element in establishing the result is identifying the right things to bound and the right bounds for them.
>
>
> — On $W_L(0)$ —
> We sample $W_L(0)$ from the uniform measure over all row-orthogonal matrices satisfying $W_L(0)W_L(0)^\top = mI_{d_y}$. Then as a consequence, in expectation we have $\mathbb{E}[W_L(0)^\top W_L(0)] = d_y I_m$. Thank you for raising the confusion, and we will explain this more clearly in the paper.
>
>
> — Experiment —
> Thank you for the suggestion! We have added a short experiment section (Section 6) to the paper. We train a family of deep linear networks with their widths ranging in [10, 1000] and depths ranging in [1, 700]. Each network is trained on the same dataset using gradient descent starting from both Gaussian and orthogonal initializations, and we produce heat-maps whose colors exhibit the losses after 10,000 steps for all configurations.
>
> The heat-maps clearly demonstrate a sharp transition from untrainable to trainable when we increase the width of the network. For Gaussian initialization, this transition occurs across a contour characterized by a linear relation between width and depth; for orthogonal initialization, the transition occurs at a width that is approximately independent of the depth. These observations match our theory excellently. Please see Section 6 for details.

---

### Decision · Program_Chairs · 2019-12-19

**Decision:**

Accept (Poster)

**Comment:**

The paper shows that initializing the parameters of a deep linear network from the orthogonal group speeds up learning, whereas sampling the parameters from a Gaussian may be harmful.

The result of this paper can be interesting to the deep learning community. The main concern the reviewers raised is the huge overlap with the paper by Du & Hu (2019). It would have been nice to actually see whether the results for linear networks empirically also hold for nonlinear networks.